# Pyocyanin and 1-Hydroxyphenazine Promote Anaerobic Killing of *Pseudomonas aeruginosa* via Single-Electron Transfer with Ferrous Iron

Jihee Kang,[a] ![ORCID] You-Hee Cho,[b] ![ORCID] Yunho Lee[a]

[a]Department of Food Science and Biotechnology, College of Life Science, CHA University, Pocheon, Gyeonggi-do, Republic of Korea
[b]Department of Pharmacy, College of Pharmacy and Institute of Pharmaceutical Sciences, CHA University, Seongnam, Gyeonggi-do, Republic of Korea

**ABSTRACT** Previously, it was reported that natural phenazines are able to support the anaerobic survival of *Pseudomonas aeruginosa* PA14 cells via electron shuttling, with electrodes poised as the terminal oxidants (Y. Wang, S. E. Kern, and D. K. Newman, J Bacteriol 192:365–369, 2010, https://doi.org/10.1128/JB.01188-09). The present study shows that both pyocyanin (PYO) and 1-hydroxyphenazine (1-OHPHZ) promoted the anaerobic killing of PA14 Δ*phz* cells presumably via a single-electron transfer reaction with ferrous iron. However, phenazine-1-carboxylic acid (PCA) did not affect anaerobic survival in the presence of ferrous iron. Anaerobic cell death was alleviated by the addition of antioxidant compounds, which inhibit electron transfer via DNA damage. Neither superoxide dismutase (SOD) nor catalase was able to alleviate *P. aeruginosa* cell death, ruling out the possibility of reactive oxygen species (ROS)-induced killing. Further, the phenazine degradation profile and the redox state-associated color changes suggested that phenazine radical intermediates are likely generated by single-electron transfer. In this study, we showed that the phenazines 1-OHPHZ and PYO anaerobically killed the cell via single-electron transfer with ferrous iron and that the killing might have resulted from phenazine radicals.

**IMPORTANCE** *Pseudomonas aeruginosa* is an opportunistic human pathogen which infects patients with burns, immunocompromised individuals, and in particular, the mucus that accumulates on the surface of the lung in cystic fibrosis (CF) patients. Phenazines as redox-active small molecules have been reported as important compounds for the control of cellular functions and virulence as well as anaerobic survival via electron shuttles. We show that both pyocyanin (PYO) and 1-hydroxyphenazine (1-OHPHZ) generate phenazine radical intermediates via presumably single-electron transfer reaction with ferrous iron, leading to the anaerobic killing of *Pseudomonas* cells. The *recA* mutant defect in the DNA repair system was more sensitive to anaerobic conditions. Our results collectively suggest that both phenazines anaerobically kill cells via DNA damage during electron transfer with iron.

**KEYWORDS** *Pseudomonas aeruginosa*, 1-hydroxyphenazine, pyocyanin, ferrous iron, anaerobic killing, phenazine radical intermediate, single-electron transfer

**Ad Hoc Peer Reviewer** ![ORCID] Michael Schurr, University of Colorado School of Medicine

Address correspondence to You-Hee Cho, youhee@cha.ac.kr, or Yunho Lee, yunho.lee@cha.ac.kr.

The authors declare no conflict of interest.

**P**seudomonas aeruginosa is a metabolically versatile Gram-negative bacterium and an opportunistic human pathogen occupying diverse niches, ranging from aquatic sediments and soils to humans (1, 2). It is best known as an opportunistic human pathogen which infects patients with burns, immunocompromised individuals, and in particular, the mucus that accumulates on the surface of the lung in cystic fibrosis (CF) patients (2). *P. aeruginosa* is capable of forming biofilms under various conditions, subsequently leading to CF lung infections, chronic wounds, and sinus infections (3–5). During biofilm infection, *P. aeruginosa* cells encounter several environmental changes,

including the depletion of oxygen and shifts in iron availability and redox chemistry (6–8). Among them, oxygen is well studied and the most familiar example (6–8). In *P. aeruginosa*, the oxygen concentration in biofilm continues to decrease with increasing depth and is depleted at the level of the biofilm (6). Thus, due to its ability to grow aerobically and anaerobically, *P. aeruginosa* is capable of forming robust anaerobic biofilms in CF mucus (9–11).

Redox-active molecules as secondary metabolites are recognized as important compounds for the control of cellular functions, such as intracellular redox homeostasis, gene expression, growth, and iron acquisition, and competition in their communities (12–17). Pseudomonads typically produce phenazines, which are redox-active small molecules with antibiotic effects. Phenazines produced by *Pseudomonas* species are mostly simple hydroxyl- and carboxyl-substituted heterocyclic compounds, and the combination of functional groups determines the physical and chemical properties, such as redox potential, affecting their biological functions and possible mode of action (18, 19). Their broad-spectrum antibiotic activity and distinctive coloration have been well characterized, with two distinct peaks in the UV range and at least one in the visible range, according to the functional groups (20). *P. aeruginosa* produces at least four different phenazines, phenazine-1-carboxylicacid (PCA), 5-*N*-methyl-1-hydroxyphenazine (pyocyanin, PYO), 1-hydroxyphenazine (1-OHPHZ), and phenazine-1-carboxamide (PCN) (21). Redox-active phenazines are best known for their antibiotic properties resulting from their ability to generate reactive oxygen species (ROS). Their toxicities presumably result from the ability to generate superoxide by donating one electron to $O_2$ (22). The anion turns into a hydroxyl radical (HO·) via hydrogen peroxide ($H_2O_2$), and the hydroxyl radical induces cell death (23, 24). PYO can generate a hydroxyl radical in the presence of Fe-pyochelin (PCH) and cause cell damage in mammalian cells (25). It is likely that both PYO and PCH can trigger induced systemic resistance (ISR) to *Botrytis cinerea* in tomato mediated by ROS (26). Due to their antibiotic activities, phenazines inhibit fungal phytopathogens (27), suppress plant diseases caused by fungi (28), and contribute to the competitiveness of other phenazine producers in the environment (12). Moreover, pyocyanin is required for virulence and acts by interfering with several cellular functions in host cells (29). In addition, pyocyanin acts as a physiological signal. Pyocyanin regulates quorum sensing-controlled genes by activating SoxR, a superoxide stress response regulator; this response is independent of oxidative stress (16, 30). Phenazines have the effect of electron transfer to terminal oxidants, such as ferric iron. This process solubilizes ferric iron, Fe(III), into bioavailable ferrous iron, Fe(II) (14). Phenazine-1-carboxylic acid promotes the development of *P. aeruginosa* biofilms by facilitating the acquisition of ferric iron (31). For these reasons, phenazines have become an active area of research in the field of environmental and clinical microbiology.

In recent studies, phenazines have mostly been recognized for their beneficial physiological role as electron shuttles under anaerobic conditions (17, 32–34). Recent studies have indicated that phenazines and redox-cycling drugs are able to directly oxidize iron-sulfur clusters of SoxR to activate them in *Escherichia coli* cells under anaerobic conditions, while superoxide does not. SoxR oxidized by these compounds then stimulates expression of the genes in the SoxRS regulon by activating the transcription of *soxS* (32). Phenazines contribute to the development of biofilms by reducing minerals to oxidants or by balancing the redox state of the cell (16, 35, 36). Under low oxygen tension and low pH conditions, reduced phenazines might be oxidized by reducing ferric iron (less soluble) from transferrin to ferrous iron (more soluble), subsequently alleviating the iron limitation of *P. aeruginosa* by ferric iron sequestration (37). In addition, phenazines can serve as terminal oxidants for balancing intracellular redox states in the depths of biofilm (19, 36). It was reported that *P. aeruginosa* can maintain intracellular redox homeostasis by reducing the phenazines in the anoxic zone of the colony biofilm. However, colony wrinkling increased the colony's surface area to access oxygen in the absence of phenazines (35). Therefore, the redox activity of phenazines might contribute to the survival of *P. aeruginosa* in the host lung during biofilm infections. In *P. aeruginosa*, phenazines (PCA, PYO, and 1-OHPHZ) promote anaerobic survival

in anaerobic reactors containing electrodes poised as phenazine-reducing potentials (33). It has been suggested that electron shuttling of phenazines, using electrodes as the terminal oxidants, might support the anaerobic survival of *P. aeruginosa*. Therefore, we hypothesized that electron shuttling of phenazines with minerals, such as iron, might promote the survival of *P. aeruginosa* in anaerobic conditions.

Here, we characterized the effect of phenazine-mediated electron transfer using ferrous iron on the survival of *P. aeruginosa* cells under anaerobic conditions. Both pyocyanin and 1-OHPHZ killed *P. aeruginosa* cells under anaerobic conditions in the presence of ferrous iron. The anaerobic killing might have been caused by DNA damage, independent of oxidative stress. Unlike anaerobic survival enhanced by the electron shuttling of phenazines using electrodes (33), one electron was transferred from ferrous iron to both oxidized PYO and 1-OHPHZ, and phenazine radical intermediates were generated under anaerobic conditions. However, PCA did not affect anaerobic killing, and the radicals were not generated in the presence of ferrous iron.

## RESULTS

**Both phenazines, 1-OHPHZ and PYO, can stimulate the anaerobic killing of *P. aeruginosa* and phenazine degradation in the presence of ferrous iron.** In their recent work, Wang, Newman, and coworkers reported that under anaerobic conditions, three natural phenazines (PCA, PYO, and 1-OHPHZ) excreted by PA14 cells can support the anaerobic survival of PA14 Δ*phz* cells by extracellular electron shuttling, with electrodes poised as the terminal oxidant, which can help phenazine electron shuttling (33). In this study, we began by measuring cell viability by adding two oxidized phenazines (1-OHPHZ and PYO, ~80 $\mu$M) to cells in the presence of ferrous iron (200 $\mu$M) to determine whether electron transfer between phenazines and ferrous iron as the terminal reductant affects the anaerobic survival of PA14 Δ*phz* cells. Unlike electrodes used as the terminal oxidant, 1-OHPHZ caused a greater than 2-log reduction in cells within 2 days under anaerobic conditions in the presence of ferrous iron, although neither phenazine nor iron affected the cell viability compared to medium alone (MOPS; morpholinepropanesulfonic acid) (Fig. 1A). In particular, 1-OHPHZ combined with ferrous iron stimulated anaerobic cell death within 6 h. We next measured the degradation of 1-OHPHZ under these conditions to see whether electron transfer would occur between the compound and ferrous iron. Degradation of 1-OHPHZ was increased up to 60.0% by ferrous iron, compared to that of 1-OHPHZ alone (20.0%), within the testing period (Fig. 1B). In addition, no other peaks were detected in any of the high-performance liquid chromatograms (HPLC) under these conditions, indicating that 1-OHPHZ was only degraded in the presence of ferrous iron (data not shown).

Based on the results, we questioned whether the pattern of anaerobic killing was dependent on the concentration of either 1-OHPHZ or ferrous iron. To answer this question, we measured the anaerobic cell viability at different concentrations of the compounds. 1-OHPHZ (80 $\mu$M) anaerobically killed cells in the presence of lower concentrations of ferrous (25 to 200 $\mu$M), and the viability of PA14 Δ*phz* cells significantly decreased within 6 h, compared to that for 1-OHPHZ alone (Fig. 1C). Moreover, the pattern of anaerobic killing was dependent on the concentration of ferrous iron. At a lower concentration of 1-OHPHZ (10 $\mu$M), the viability of PA14 Δ*phz* cells significantly decreased within 6 h by the application of more than 100 $\mu$M ferrous iron, but lower concentrations of ferrous iron (10 to 50 $\mu$M) did not affect anaerobic cell viability (Fig. 1D). These results indicate that 1-OHPHZ in combination with ferrous iron kills PA14 Δ*phz* cells very quickly under anaerobic conditions at rates that are correlated with the concentrations of both compounds.

To investigate whether another phenazine would have the same effect as 1-OHPHZ, we measured anaerobic cell viability by adding PYO and ferrous iron. Similarly to 1-OHPHZ, PYO also stimulated anaerobic PA14 Δ*phz* cell death in the presence of ferrous iron within 6 h, although PYO killed fewer cells than 1-OHPHZ in the presence of ferrous iron (Fig. 2A). PA14 Δ*phz* cells also degraded ~18.0% of the PYO in the presence of ferrous iron within 6 h, but only 3.0% of PYO was degraded in the absence of ferrous

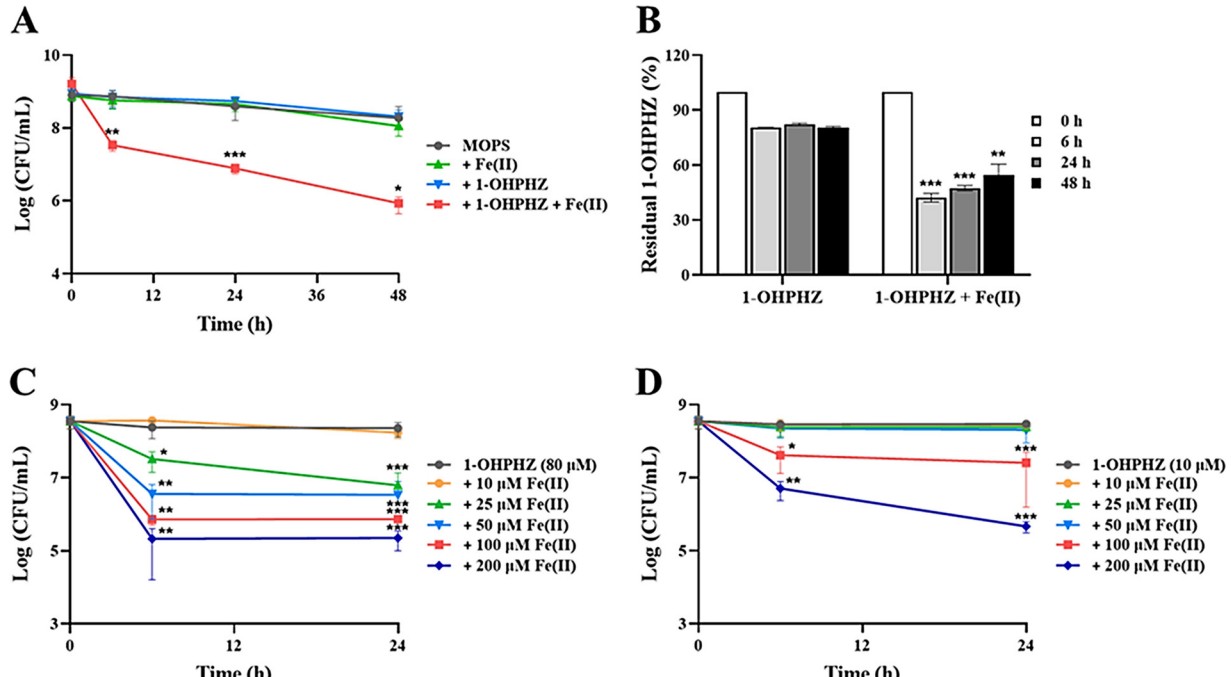

**FIG 1** Anaerobic killing of PA14 Δ*phz* cells by 1-OHPHZ via electron transfer with ferrous iron. (A) The phenazine 1-OHPHZ (80 μM) promoted anaerobic killing of the Δ*phz* mutant of *P. aeruginosa* PA14 via electron transfer with ferrous iron (200 μM) and (B) led to phenazine degradation when the cells were anaerobically incubated at 30°C with agitation (225 rpm) in MOPS minimal medium. Incubations with ferrous iron alone, 1-OHPHZ alone, and with neither served as the controls. The phenazine concentrations in culture filtrates were determined at each time point. (C and D) Both concentrations of 1-OHPHZ—80 μM (C) and 10 μM (D)—promoted the anaerobic killing of PA14 Δ*phz* cells incubated in MOPS minimal medium supplemented with a wide range of ferrous iron concentrations (10 to 200 μM). The error bars represent the standard deviation from two independent experiments. Significance was calculated by one-way analysis of variance (ANOVA) with Bonferroni's multiple-comparison test. *, $P < 0.05$; **, $P < 0.01$; ***, $P < 0.001$.

iron (Fig. 2B). Just as PYO caused anaerobic cell death at a lower rate than that of 1-OHPHZ, the rate of degradation of PYO was also smaller than that of 1-OHPHZ in the presence of ferrous iron (Fig. 2B). These data suggest that the phenazines 1-OHPHZ and PYO stimulate anaerobic cell death of *P. aeruginosa* by electron transfer with ferrous iron and that the anaerobic killing is correlated with degradation of the phenazines.

Even if phenazines generate superoxide from $O_2$ under aerobic conditions (22), we questioned whether the anaerobic killing was caused by reactive oxygen species (ROS) which was generated by the phenazines from residual oxygen inside the cells or by ferrous iron via Fenton reaction inside the cells (24). To answer this question, we used

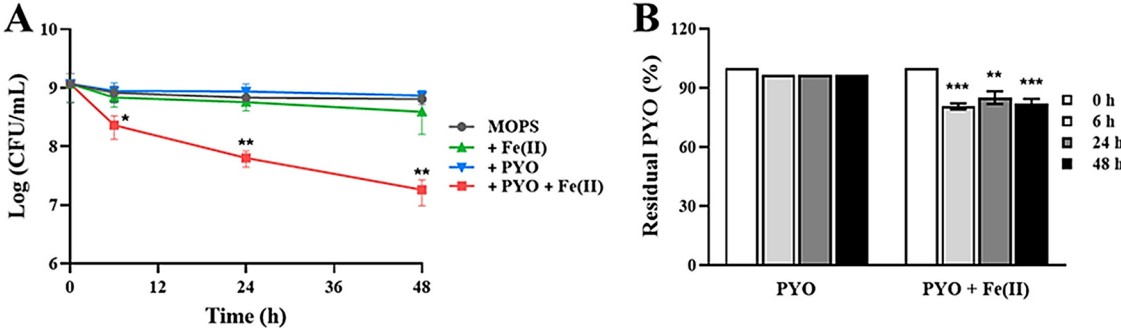

**FIG 2** Anaerobic killing of PA14 Δ*phz* cells by PYO via electron transfer with ferrous iron. (A) The phenazine PYO (80 μM) in combination with ferrous iron (200 μM) promotes anaerobic killing of PA14 Δ*phz* cells incubated at 30°C with agitation (225 rpm) in MOPS medium. Incubation with ferrous iron alone, with PYO alone, or with neither served as the controls. (B) Phenazine concentrations in culture filtrates were determined at each time point. The error bars represent the standard deviation from two independent experiments. Significance was calculated by one-way analysis of variance (ANOVA) with Bonferroni's multiple-comparison test. *, $P < 0.05$; **, $P < 0.01$; ***, $P < 0.001$.

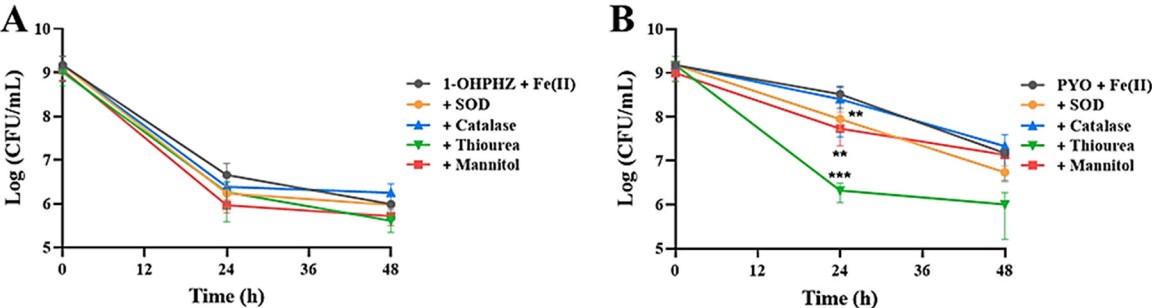

**FIG 3** Effects of ROS scavengers on anaerobic killing of PA14 Δ*phz* cells by the phenazines 1-OHPHZ (A) and PYO (B) via electron transfer with ferrous iron. None of the ROS scavengers tested in this study affected the anaerobic killing of PA14 Δ*phz* cells by the phenazines (80 $\mu$M) 1-OHPHZ or PYO combined with ferrous iron (200 $\mu$M) when the cells were anaerobically incubated at 30°C with agitation (225 rpm) in MOPS minimal medium. The error bars represent the standard deviation from two independent experiments. Significance was calculated by one-way analysis of variance (ANOVA) with Bonferroni's multiple-comparison test. *, $P < 0.05$; **, $P < 0.01$; ***, $P < 0.001$. Superoxide dismutase (SOD), 5,000 U/mL; catalase, 2,500 U/mL; thiourea, 150 mM; mannitol, 2.0 mM.

four different ROS scavengers, superoxide dismutase (SOD), catalase, thiourea, and mannitol, which can detoxify $O_2^-$, $H_2O_2$, $HO^.$, and $HO^.$, respectively (23, 38). As indicated by Fig. 3, not all scavengers tested in this study affected cell viability by electron transfer between the phenazines (1-OHPHZ and PYO) and ferrous iron. Rather, the scavengers SOD, thiourea, and mannitol induced the anaerobic killing by PYO together with ferrous iron at 24 h (Fig. 3B). The data showed that ROS were not generated by electron transfer under our anaerobic conditions, and the mechanism of anaerobic killing was independent of oxidative stress. Redox-active phenazines are capable of generating superoxide and hydrogen peroxide from $O_2$ as a terminal oxidant under aerobic conditions (22). Hydroxyl radicals are generated from hydrogen peroxide by ferrous iron via Fenton reaction in the cell (23, 24). For resistance toward the toxicity of self-produced phenazines, *P. aeruginosa* seems to have at least two different mechanisms, and it was noted that it exhibited increased superoxide dismutase and catalase activity under conditions favoring pyocyanin production (39). Therefore, we tested the effect of ROS scavengers, superoxide dismutase and catalase, to determine whether the anaerobic killing was caused by ROS. However, these reactive oxygen species were not involved in anaerobic killing in experiments with the scavengers (Fig. 3).

**Antioxidants alleviate anaerobic killing by electron transfer between the phenazines 1-OHPHZ and PYO and ferrous iron.** In Fig. 1 and 2, the phenazines 1-OHPHZ and PYO combined with ferrous iron stimulated anaerobic killing of PA14 Δ*phz* cells and their degradation, and the data strongly suggests that the phenazines stimulated anaerobic killing via electron transfer with ferrous iron. To confirm that, we tested whether the viability would be rescued by adding 2,2'-dipyridyl (DP), as a ferrous iron chelator, to remove ferrous iron from the medium (23). As expected, the anaerobic killing by both 1-OHPHZ (Fig. 4A) and PYO (Fig. 4C) in the presence of ferrous iron was rescued by the application of 2,2'-dipyridyl (1.0 mM) to the cultures, which inhibited electron transfer between the phenazines and ferrous iron. We measured the phenazine concentrations with the addition of DP because the anaerobic killing correlated with phenazine degradation. As with the anaerobic killing results, the degradation of phenazine was reduced when DP was added (Fig. 4B and D). In addition, we also measured anaerobic killing in the presence of other antioxidants, such as dithionite (DTT) and glutathione (GSH). These biological (GSH) and artificial (DTT) electron donors have been known to catalytically reduce hydrogen peroxide or redox-cycling drugs (32, 40). Both antioxidants also rescued anaerobic cell viability from the killed by 1-OHPHZ (Fig. 5A) or PYO (Fig. 5B) in combination with ferrous iron when the cell was incubated over 2 days. These antioxidants, which inhibit electron transfer between the phenazines and ferrous iron by either removing ferrous iron or reducing phenazine, alleviated anaerobic killing. Therefore, the results indicate that the phenazines 1-OHPHZ and PYO anaerobically kill cells via electron transfer with ferrous iron.

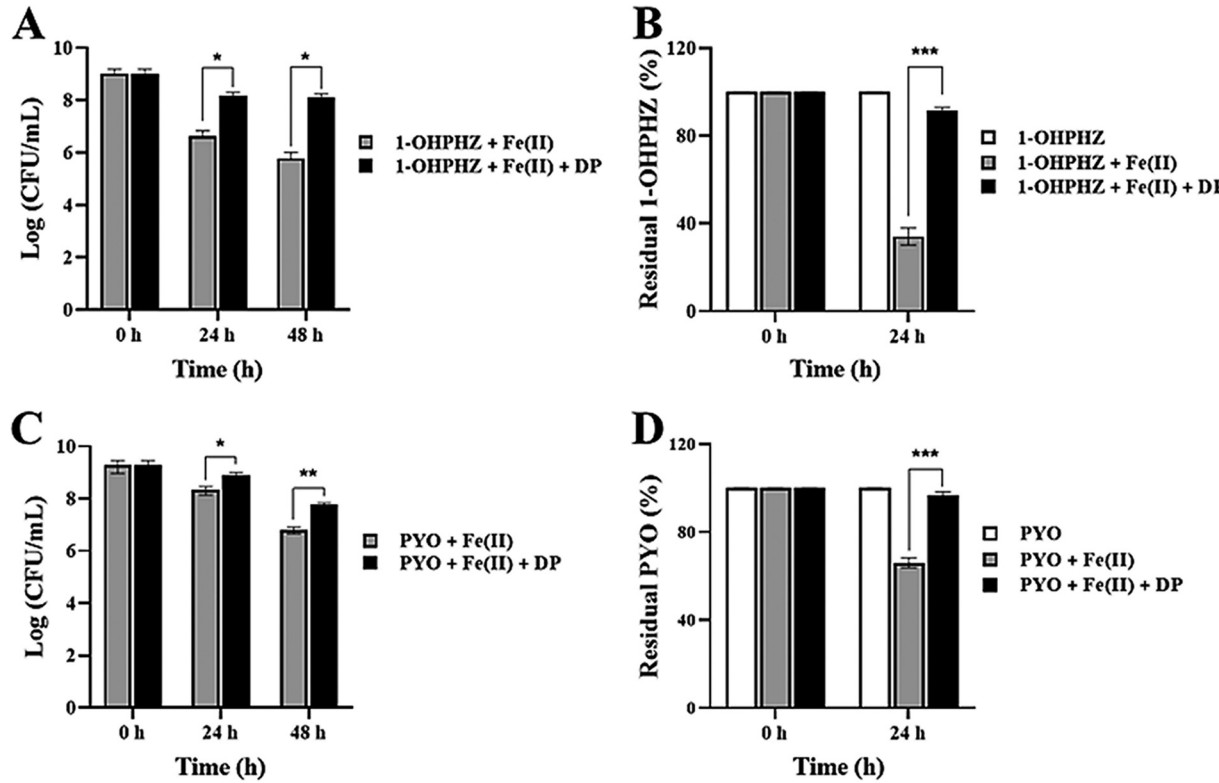

**FIG 4** Effects of 2,2′-dipyridyl (DP) on anaerobic killing of PA14 Δ*phz* cells by phenazines PYO and 1-OHPHZ via electron transfer with ferrous iron. DP (1.0 mM) rescued PA14 Δ*phz* cells from anaerobic killing by the phenazines (80 $\mu$M) 1-OHPHZ (A) and PYO (C) combined with ferrous iron (200 $\mu$M) and inhibited degradation of the phenazines 1-OHPHZ (B) and PYO (D) when the cells were anaerobically incubated at 30°C with agitation (225 rpm) in MOPS minimal medium. The phenazine concentrations in the culture filtrates were determined at each time point. The error bars represent the standard deviation from two independent experiments. Significance was calculated by one-way analysis of variance (ANOVA) with Bonferroni's multiple-comparison test. *, $P < 0.05$; **, $P < 0.01$; ***, $P < 0.001$.

**Phenazine radical intermediates are anaerobically generated by electron transfer with ferrous iron.** To verify electron transfer between the phenazines and ferrous iron, we determined the redox states of phenazines by observation of their colors. Phenazines are redox-active compounds that change their colors depending on the pH value and oxidation states. At a neutral pH, PYO is bright blue, and it becomes colorless by two-electron reduction. In addition, when oxidized 1-OHPHZ is fully reduced, its color turns from orange to colorless (19, 22). Therefore, the characteristic color and

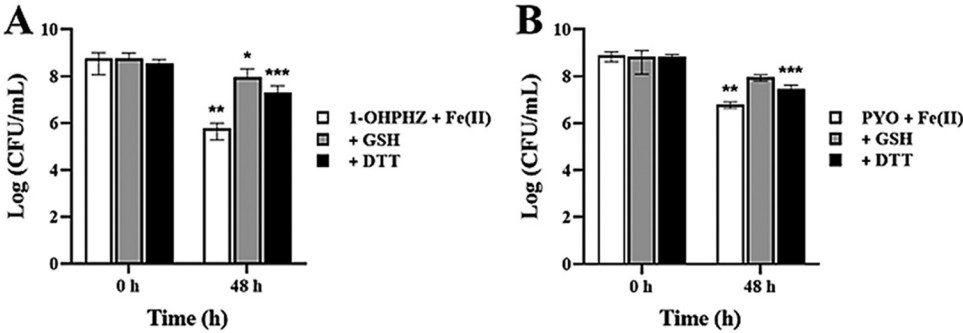

**FIG 5** Effects of antioxidants on anaerobic killing of PA14 Δ*phz* cells by the phenazines PYO and 1-OHPHZ, combined with ferrous iron. Both dithionite (DTT; 1.0 mM) and glutathione (GSH; 1.0 mM) rescued PA14 Δ*phz* cells from anaerobic killing by the phenazines (80 $\mu$M) 1-OHPHZ (A) and PYO (B) combined with ferrous iron (200 $\mu$M) when the cells were anaerobically incubated at 30°C with agitation (225 rpm) in MOPS minimal medium. The error bars represent the standard deviation from two independent experiments. Significance was calculated by one-way analysis of variance (ANOVA) with Bonferroni's multiple-comparison test. *, $P < 0.05$; **, $P < 0.01$; ***, $P < 0.001$.

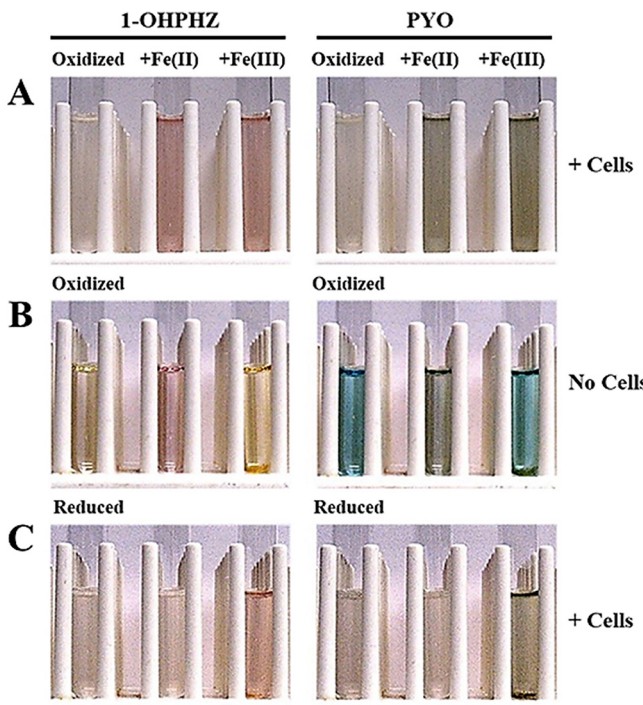

**FIG 6** Color changes of phenazines caused by ferrous iron via single-electron transfer. (A) Color changes of phenazines, 1-OHPHZ (left panel) and PYO (right panel), with the addition of 200 $\mu$M ferrous iron and ferric iron after PA14 $\Delta phz$ cells anaerobically reduced the oxidized phenazines for 24 h, compared to the control (C) (no iron added). (B) Ferrous iron changed the colors of both oxidized 1-OHPHZ (left panel) and PYO (right panel) in the absence of cells. (C) Color changes of both reduced 1-OHPHZ (left panel) and PYO (right panel) were caused by the addition of ferric iron after the cells completely reduced oxidized phenazines for 24 h.

absorption spectrum of phenazines indicate their redox state. First, we observed how their colors were changed by ferrous iron when the cell was anaerobically incubated in MOPS minimal medium supplemented with both oxidized phenazines and ferrous iron for 24 h (Fig. 6). The phenazines turned colorless in the absence of ferrous iron because they were fully reduced by the cell. *P. aeruginosa* pyocyanin can easily penetrate cell membranes due to its zwitterionic properties, and it is reduced by directly accepting two electrons (refer to electron shuttling) from NADH or reduced glutathione in the cell (41, 42). Therefore, when the cells were grown in MOPS medium with the oxidized phenazine alone, they turned colorless under anaerobic conditions. However, their colors turned pink (1-OHPHZ) or green (PYO) in the presence of ferrous iron, indicating that ferrous iron affects the reduction of phenazines by cellular NADH or glutathione (Fig. 6A). Furthermore, we tested the effects of iron on the redox states of oxidized/reduced phenazines separately to verify which phenazine redox state was affected by ferrous iron. We observed that the colors of the oxidized phenazines were changed by ferrous iron in the absence of cells because the oxidized compounds were reduced by the cells. The colors of oxidized 1-OHPHZ (orange) and PYO (blue) immediately turned pink and green, respectively, when ferrous iron was added (Fig. 6B). The changed colors are easily distinguished from those of the oxidized phenazines themselves, as well as those of the reduced compounds (colorless). 1-OHPHZ was degraded about 28.0% when oxidized 1-OHPHZ was transformed into the radical intermediate by ferrous iron. In addition, it maintained the colors of radical intermediates for around 2 days, suggesting that its life span was extended (data not shown). We next observed how the redox states of reduced phenazines were changed by ferrous iron. To ensure they were fully reduced, we grew the cell anaerobically in MOPS medium with phenazine alone for 24 h and then added ferrous iron to the cultures. However, the colors of reduced 1-OHPHZ (colorless) and PYO (colorless) were not changed by ferrous iron (Fig. 6C).

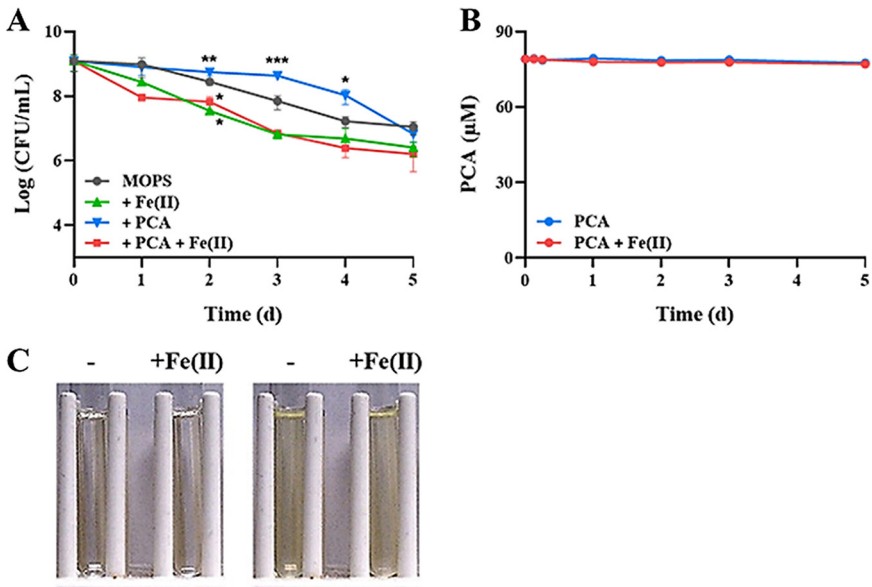

**FIG 7** Anaerobic viability of PA14 Δ*phz* cells with the phenazine PCA and ferrous iron and degradation of PCA. (A) PCA (80 $\mu$M) with ferrous iron (200 $\mu$M) did not affect the anaerobic viability of PA14 Δ*phz* cells anaerobically incubated at 30°C with agitation (225 rpm) in MOPS minimal medium. (B) PCA concentrations remained unchanged, indicating that it did not degrade. Incubations with ferrous iron alone, with PCA alone, and with neither served as the controls. The numbers of viable cells and concentrations of PCA in the culture filtrates were determined over time. (C) Ferrous iron did not affect the redox states of either oxidized (left panel) or reduced (right panel) PCA. The error bars represent the standard deviation from two independent experiments. Significance was calculated by one-way analysis of variance (ANOVA) with Bonferroni's multiple-comparison test. *, $P < 0.05$; **, $P < 0.01$; ***, $P < 0.001$.

From these results, we hypothesized that ferric iron could affect the redox states of phenazines, generating the radical intermediates, similarly to ferrous iron. As expected, ferric iron, as a terminal oxidant, partially oxidized reduced 1-OHPHZ (colorless to pink) or PYO (colorless to green), although ferrous iron did not affect the compounds (Fig. 6). This result suggested that oxidized 1-OHPHZ and PYO were partially reduced by ferrous iron as a terminal reductant without enzymatic catalysis and that the phenazines were transformed into radical intermediates by presumably single-electron transfer with ferrous iron, not electron shuttling, under anaerobic conditions. Moreover, this transformation would then stimulate phenazine degradation because radical intermediates are unstable and their half-time is short. We hypothesize that the radicals contributed to anaerobic killing. However, this remains to be further characterized.

**PCA cannot stimulate anaerobic killing or degradation by electron transfer with ferrous iron.** To confirm our observation and hypothesis, we also measured how the viability of PA14 Δ*phz* cells under anaerobic conditions was affected by the other phenazine, PCA, as well as whether PCA would degrade. Unlike both 1-OHPHZ and PYO, PCA did not affect anaerobic cell viability in the presence of ferrous iron over 5 days, compared to controls (Fig. 7A). Therefore, we expected that PCA would not degrade in the presence of ferrous iron because the anaerobic killing was correlated with phenazine degradation. As we expected, PCA did not degrade at all under anaerobic conditions (Fig. 7B). In addition, the color of the oxidized/reduced PCA was not changed by ferrous iron (Fig. 7C). These data suggest that both 1-OHPHZ and PYO anaerobically kill cells via electron transfer with ferrous iron and that the process (single-electron transfer) is different from that of PCA. The data suggest that a single electron of ferrous iron is transferred to oxidized PYO or 1-OHPHZ rather than a 2-electron reduction, due to their different reactivity with ferrous iron from that of PCA, generating the intermediates under anaerobic conditions.

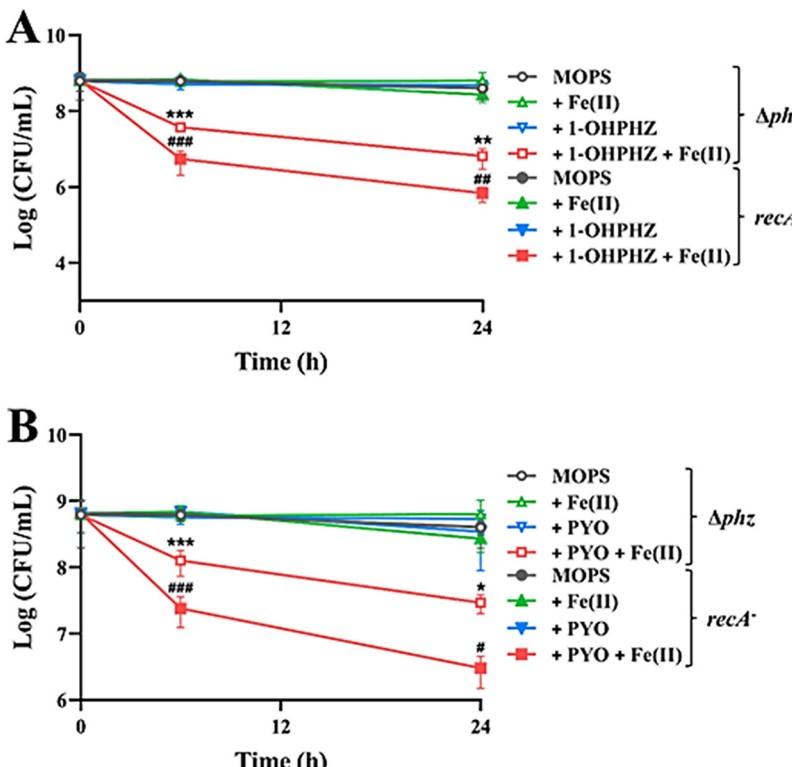

**FIG 8** Anaerobic killing of *recA* mutant cells by the phenazines 1-OHPHZ (A) and PYO (B) via single-electron transfer with ferrous iron. *recA* mutant cells exhibited greater sensitivity to electron transfer between 1-OHPHZ and PYO (80 $\mu$M) and ferrous iron (200 $\mu$M) than PA14 $\Delta phz$ cells. The error bars represent the standard deviation from two independent experiments. Significance was calculated by one-way analysis of variance (ANOVA) with Bonferroni's multiple-comparison test. *, $P < 0.05$; **, $P < 0.01$; ***, $P < 0.001$.

**The *recA* mutant defect in the DNA repair system is more sensitive to electron transfer between the phenazines and ferrous iron.** In recent studies, Gates's group reported that myxin, which is a phenazine produced by *Sorangium* strains, causes radical-mediated DNA strand cleavage under both aerobic and anaerobic conditions (38). To verify whether electron transfer between the phenazines and ferrous iron would stimulate anaerobic killing via DNA damage, we measured the cell viability of the *recA* mutant (43), which lacks a DNA repair system (44). The parental strain of the mutant is *P. aeruginosa* PA14 wild type (43). PA14 wild type produces PCA and subsequently modifies PCA to PYO under aerobic conditions. However, PA14 wild type cannot synthesize PYO, although it produces PCA under anaerobic conditions (14). Although the *recA* mutant produces PCA, PCA did not affect the anaerobic cell viability in the absence/presence of ferrous iron (Fig. 7A). Therefore, the *recA* mutant was aerobically grown in MOPS minimal medium and washed thoroughly with MOPS basal medium to remove residual PYO from the medium. Then, the viabilities of *recA* mutant and PA14 $\Delta phz$ cells grown in MOPS with phenazines and iron were determined under anaerobic conditions. Neither chemical affected the viability of the *recA* mutant cells, compared to their viability in MOPS medium. The *recA* mutant cells exhibited greater sensitivity to the anaerobic killing conditions than the PA14 $\Delta phz$ cells (Fig. 8). In particular, the viability of the *recA* mutant decreased by more than 1 log in 1-OHPHZ (Fig. 8A) or PYO (Fig. 8B) combined with ferrous iron. These data imply that radical intermediates generated by single-electron transfer between the phenazines and ferrous iron cause DNA damage and directly or indirectly enhance anaerobic killing. In addition, PYO caused more DNA damage than 1-OHPHZ in the presence of ferrous iron.

## DISCUSSION

We showed that three phenazines differentially affect the viability of PA14 $\Delta phz$ cells via electron transfer with ferrous iron as a terminal reductant under anaerobic conditions.

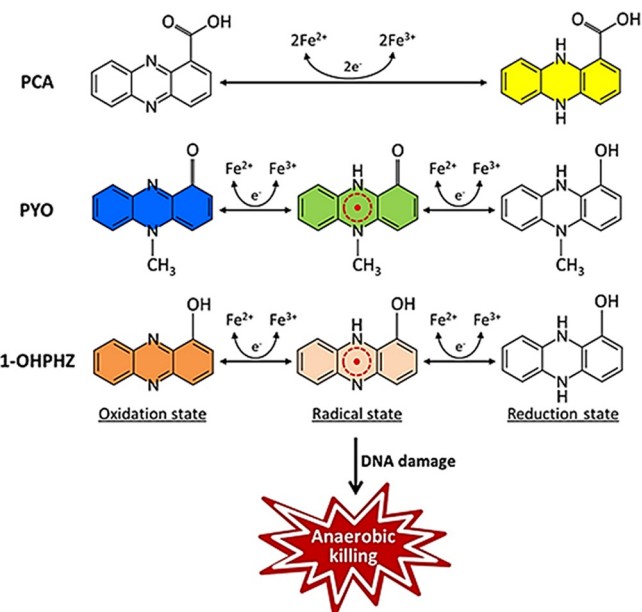

**FIG 9** Proposed model of function of electron transfer between three natural phenazines and iron and its impact on survival of *P. aeruginosa* cells under anaerobic conditions. The phenazines 1-OHPHZ and PYO are partially reduced by single-electron transfer with ferrous iron outside the inner membrane of *P. aeruginosa* cells, generating radical intermediates. The intermediates stimulate anaerobic killing, mediated indirectly or directly by DNA damage.

Of the phenazines, 1-OHPHZ and PYO significantly stimulated anaerobic killing of PA14 Δ*phz* cells in the presence of ferrous iron. The anaerobic killing exhibited a correlation with phenazine degradation as well as concentrations of both phenazine and ferrous iron (Fig. 1 and 2). However, the other phenazine, PCA, did not affect the anaerobic survival of PA14 Δ*phz* cells, and it was not degraded in the presence of ferrous iron under the same conditions (Fig. 7). A recent paper reported that three phenazines, PCA, PYO, and 1-OHPHZ, can support the survival of *P. aeruginosa* cells in anaerobic reactors containing electrodes poised as phenazine-reducing potentials (33). Phenazines have also been known to require two electrons to be fully reduced or oxidized and are capable of undergoing reduction by NAD(P)H in the cells and oxidation by terminal oxidants such as minerals and molecular oxygen (19, 36). Therefore, in anaerobic reactors, reduced phenazines were fully reoxidized by electrodes poised outside the cells, and the process may continue until the phenazines are degraded (33). This finding suggests that two-electron reduction/oxidation of phenazines (electron shuttling) might support anaerobic survival. However, the mechanisms still remain to be characterized (33). In our study, ferrous iron partially reduced fully oxidized 1-OHPHZ and PYO, not electron shuttling, and subsequently, phenazine radical intermediates were generated (Fig. 6). This implies that phenazines can serve as the terminal oxidant instead of oxygen under anaerobic conditions. Moreover, phenazine electron shuttling alters intracellular redox states and facilitates biofilm formation, promoting anaerobic survival (15, 16, 33). In addition, PCA combined with ferrous iron did not support anaerobic survival in our tested conditions, although we predicted that electron shuttling would occur between PCA and ferrous iron (33). We estimated that redox cycling of PCA (80 $\mu$M) might occur fewer than 3 times in the presence of ferrous iron (200 $\mu$M), and the amount of cycling is too short to support anaerobic survival. Similarly, we previously observed that PCA can support anaerobic survival at high concentrations of ferric iron (~10 mM) (unpublished data). Therefore, radical intermediates generated by single-electron transfer with ferrous iron might imbalance the redox state, promoting anaerobic killing (Fig. 9).

We also hypothesized that an imbalanced redox state might occur in either the periplasmic membrane or outer membrane, because phenazines can be reduced by NAD(P)H or reduced glutathione in cytoplasm (41, 42). To find the target location, we

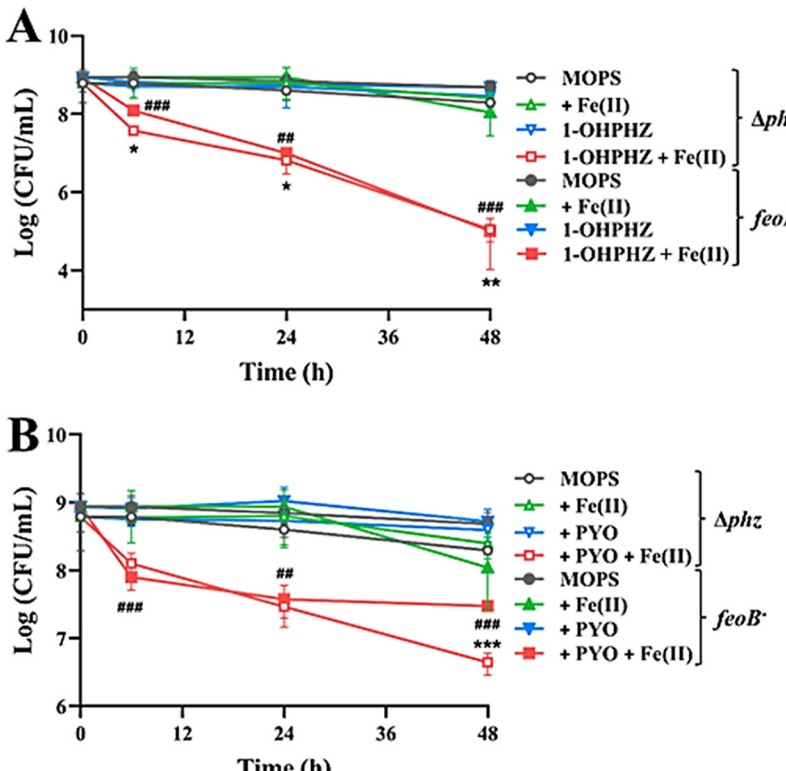

**FIG 10** Anaerobic killing of *feoB* mutant cells by the phenazines 1-OHPHZ (A) and PYO (B) via single-electron transfer with ferrous iron. There was no difference between the anaerobic killing of *feoB* mutant and PA14 Δ*phz* cells by electron transfer between 1-OHPHZ and PYO (80 $\mu$M) and ferrous iron (200 $\mu$M). The error bars represent the standard deviation from two independent experiments. Significance was calculated by one-way analysis of variance (ANOVA) with Bonferroni's multiple-comparison test. *, $P < 0.05$; **, $P < 0.01$; ***, $P < 0.001$.

tested the anaerobic killing of mutant cells with *feoB* disrupted (43); this protein is located in the inner membrane, and in the mutant, the entrance of ferrous iron through this location is blocked (45). As we expected, there was no difference in viability between *feoB* mutant cells and PA14 Δ*phz* cells (Fig. 10), which indicates that the radical intermediates were generated outside the cytoplasmic membrane, such as either periplasmic or outer membranes. Interestingly, the *recA* mutant cells exhibited greater sensitivity to the anaerobic killing conditions than the PA14 Δ*phz* cells (Fig. 8).

In a recent publication, Gates's group reported that myxin, which is a phenazine produced by *Sorangium* strains, causes radical-mediated DNA strand cleavage under both aerobic and anaerobic conditions (38). Myxin is a class of heterocyclic *N*-oxides which generate oxygen-sensitive radical intermediates by intracellular single-electron reduction. Under anaerobic conditions, myxin radical intermediates cause DNA damage via a "deoxygenative" mechanism involving the loss of oxygen from their *N*-oxide groups. Although it is not clear whether this mechanism occurs in 1-OHPHZ and PYO under anaerobic conditions, it suggests that both phenazines cause anaerobic cell death via DNA damage during electron transfer with iron. However, this mechanism needs to be further elucidated.

Redox-active phenazines function by altering intracellular redox states and facilitating both competition with other bacteria and biofilm formation on the surfaces of hosts, ranging from plants to humans (12, 15, 29, 35). In particular, concentrations of PYO of up to 100 $\mu$M are often detected in the sputa of *P. aeruginosa*-infected CF patients, and it has been known to mediate virulence (46). However, our observations reveal that phenazines may kill the cell itself in the presence of ferrous iron in the anoxic zone of colony biofilm on the lungs of CF patients.

## MATERIALS AND METHODS

**Chemicals used in this study.** Phenazine-1-carboxylic acid (PCA), pyocyanin (PYO), and 1-hydroxyphenazine (1-OHPHZ) were purchased from Princeton BioMolecular Research, Cayman Chemical Company, and TCI America, respectively. Trifluoroacetic acid (TFA), gentamicin solution, and $MgSO_4 \cdot 7H_2O$ were purchased from Sigma-Aldrich. D-glucose monohydrate and $KH_2PO_4$ were purchased from Alfa Aesar. Luria-Bertani (LB) dehydrated medium, $NaNO_3$, yeast extract, and agar powder were purchased from EMD Millipore. Acetonitrile, NaOH, and NaCl were purchased from BD Chemicals. Anhydrous dextrose and KCl were purchased from VWR Chemicals BDH. All other chemicals were purchased from Sigma-Aldrich unless otherwise stated.

**Bacterial strains and growth conditions.** In this study, we used a phenazine-null mutant of *P. aeruginosa* PA14 ($\Delta phz$), with both phenazine biosynthetic operons deleted (*phzA1-phzG1* and *phzA2-phzG2*) so that it was not capable of producing phenazines, to investigate the effects of phenazines on anaerobic survival (30). The *recA* and *feoB* mutants were obtained from the Harvard Medical School (43). *P. aeruginosa* strains were grown routinely at 37°C with vigorous shaking at 225 rpm in liquid LB medium or MOPS-based medium (100 mM MOPS at pH 7.2, 93 mM $NH_4Cl$, 43 mM NaCl, 0.22 mM $KH_2PO_4$, 1 mM $MgSO_4$, 3.6 $\mu$M $FeCl_3$, and 10 mM glucose) using a modification of a previously described method (47). Gentamicin (15 $\mu$g/mL) was used for the *recA* and *feoB* mutants.

**Anaerobic survival.** The anaerobic survival assay was performed as previously described (33). *P. aeruginosa* strains were grown overnight in liquid LB medium. The overnight cultures were inoculated with 250-fold dilutions in MOPS-based medium and grown aerobically. At the early stationary phase (at an optical density at 500 nm [$OD_{500}$] of around 1.0), the cells were harvested, washed with MOPS-based medium, and concentrated. The cells were reinoculated to obtain a cell density of $10^9$ CFU/mL into anoxic MOPS basal medium in an anaerobic chamber containing a nitrogen gas ($N_2$, 100%) (MBRAUN). Prior to inoculation, water, ferrous iron [$Fe(NH_4)_2(SO_4)_2$], ferric iron ($FeCl_3$), or oxidized phenazines (PCA, PYO, or 1-OHPHZ; 80 $\mu$M) were added to the anoxic MOPS basal medium. When necessary, ROS scavengers (superoxide dismutase, catalase, thiourea, and mannitol) or antioxidants [dithiothreitol (DTT), glutathione (GSH), and 2,2'-dipyridyl (DP)] at various concentrations were added to the anoxic cultures, containing oxidized phenazine (PYO or 1-OHPHZ) as well as ferrous iron. The cultures were anaerobically incubated in rubber-stoppered tubes at 30°C with vigorous shaking. At designated time points, 0.2 mL aliquots of culture were removed by syringe and catalase (200 U) was added to remove $H_2O_2$ generated from residual oxygen inside the cells (48). Serially diluted suspensions were anaerobically spotted onto LB plates supplemented with potassium nitrate ($KNO_3$; 100 mM) for anaerobic growth (9). Viable cells were determined by counting the colony-forming units (CFU).

**Phenazine analysis.** For analyzing phenazine degradation, the cultures were grown as described above, and the samples were prepared in 3 steps to quantify the total phenazines. (i) Anaerobic cell culture samples were immediately exposed to air to efficiently oxidize the reduced phenazines, because oxidized phenazines can be predominantly released into the culture, unlike reduced forms (41, 42). Cell-free supernatants, which comprised phenazines released into medium, were obtained by centrifugation at 13,200 rpm for 5 min. (ii) Subsequently, the pellets were resuspended in fresh medium, and supernatants were obtained to quantify the phenazines adhering to the cellular membrane. (iii) Finally, the pellets were again resuspend in fresh medium and frozen/thawed three times (dry ice to 37°C water bath) to detect phenazine remaining inside the cells by lysing them. The samples were directly loaded onto a Waters reverse-phase HPLC with a diode array UV-visible (UV-Vis) light detector and a Waters XBridge $C_{18}$ analytical column (particle size, 3.5 $\mu$m; 4.6 by 150 mm). Analysis was performed in a gradient of water–0.1% formic acid (solvent A) to acetonitrile–0.1% formic acid (solvent B) at a flow rate of 1.0 mL/min using the following method. For 0 to 10 min, chromatography was in a linear gradient from 0% solvent B to 5% solvent B; for 10 to 25 min, chromatography was in a linear gradient from 5% solvent B to 95% solvent B; for 25 to 30 min, in 95% solvent B; for 30 to 30.1 min, in a sharp linear gradient to 100% solvent A, for 30.1 to 40 min, in 100% solvent A. PCA, PYO, and 1-OHPHZ were eluted at retention times of 20.6, 14.5, and 20.1 min, respectively.

**Statistical analysis.** All experiments described above were performed at least in triplicate, with three to six biological replicates in each. Data are depicted as the mean plus or minus the standard error of the mean. Statistical analyses were performed using GraphPad Prism 9 (GraphPad Software, USA). Independent sample *t* tests and one-way ANOVA were used to detect statistically significant differences. *P* values of 0.05 or less were considered to be statistically significant.

**Data availability.** The data generated during the current study are available from the corresponding author upon reasonable request.

## ACKNOWLEDGMENTS

This research was supported by National Research Foundation of Korea (NRF) grants funded by the Ministry of Education (NRF-2018R1D1A1A02085563) and the Ministry of Science and ICT (NRF-2022R1A2C3003943), as well as by the Industry Academic Cooperation Foundation Fund, CHA University Grant (CHA-201900420001).

We sincerely thank Yun Wang and her team for helpful discussions.

J.K. and Y.L. conceived the study, performed the experimental work, analyzed the data, and wrote the paper. Y.-H.C. commented on the revision strategy and wrote the paper.

We declare no potential conflicts of interest with respect to the research, authorship, and/or publication of this article.

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
