## [Reviewer comments · Microbiology Spectrum]

Microbiology Spectrum

Pyocyanin and 1-hydroxyphenazine promote anaerobic killing of *Pseudomonas aeruginosa* via single-electron transfer with ferrous iron

Jihee Kang, Yunho Lee, and You-Hee Cho

Corresponding Author(s): Yunho Lee, CHA University

Review Timeline:

Submission Date:	June 20, 2022
Editorial Decision:	August 3, 2022
Revision Received:	September 16, 2022
Accepted:	October 17, 2022

Editor: Neha Garg

Reviewer(s): Disclosure of reviewer identity is with reference to reviewer comments included in decision letter(s). The following individuals involved in review of your submission have agreed to reveal their identity: Michael J Schurr (Reviewer #2)

Transaction Report:

DOI: <https://doi.org/10.1128/spectrum.02312-22>

August 3, 2022

Prof. Yunho Lee
CHA University
Food Science and Biotechnology
Pocheon, Gyeonggi-do 11160
Korea (South), Republic of

Re: Spectrum02312-22 (Pyocyanin and 1-hydroxyphenazine promote anaerobic killing of *Pseudomonas aeruginosa* via single-electron transfer with ferrous iron)

Dear Prof. Yunho Lee:

Link Not Available

Sincerely,

Neha Garg

Journals Department
Reviewer comments:

Reviewer #1 (Comments for the Author):

See below

Reviewer #2 (Comments for the Author):

The authors describe the ability of 1-OHPHZ and PYO to kill *P. aeruginosa* delta phz and provide evidence that these molecules are oxidized by ferrous iron that in turn damages bacterial DNA. The data support the conclusions in general.

1. Figure 6 of the manuscript is described vaguely and the figure needs additional labeling to follow. Is there another way to demonstrate the oxidized forms of these phenazines? mass spectroscopy?
2. Does ferrous iron transfer electrons to 1-OHPHZ and PYO in other media? or are these observations specific to MOPS minimal medium?
3. DNA damage can be measured. Some other assay is needed to confirm that DNA damage has occurred in the PYO and/or 1-OHPHZ and Fe(II) exposed cells besides CFU decrease.

Minor issues:

1. pg. 2, lines 46-48. The sentence starting with "Otherwise," needs to be rewritten.
2. pg. 2, line 38 "Human" should be "humans".
3. pg. 4, line 105 "transferred of" should be "transferred from".
4. pg. 4, line 111. "In recent" should be "in recent work".
5. pg 4, line 123. "whether it happens" should be "whether electron transfer between the compound and ferrous iron happens".
6. pg. 5, line 63. "ability of generation" should be "ability to generate superoxide".
7. pg. 3, line 65. "PYO can generate hydroxyl" should be "PYO can generate a hydroxyl".
8. pg. 3, line 71. "from the reasons" should be "For these reasons" and moved to the end of the sentence.
9. pg. 3, line 80. "in recent" should be "In recent studies".
10. pg. 3, line 90. "It reported" should be "It is reported".
11. pg. 5, lines 125-126. The sentence starting with "In addition" needs to be rewritten.
12. pg. 6, line 164. "and catalase whether" should be "catalase to determine whether".
13. pg. 6, lines 175-178. Two Sentences starting with "We next" need to be rewritten.
14. pg. 6, line 182. Delete "The", capitalize "B" in both.
15. pg. 7, line 201. "cell was grown" should be "cells were grown".
16. Figure 6 needs additional labeling to follow the Results on page 7 of the text.
17. pg. 8, line 234. needs to be rewritten.
18. pg. 8, line 242. "In recent" should be "In recent studies".

Staff Comments:

Preparing Revision Guidelines

Please return the manuscript within 60 days; if you cannot complete the modification within this time period, please contact me. If you do not wish to modify the manuscript and prefer to submit it to another journal, please notify me of your decision immediately so that the manuscript may be formally withdrawn from consideration by Microbiology Spectrum.

Ref: Spectrum02312-22

Dear Prof. Yunho Lee:

<https://spectrum.msubmit.net/cgi-bin/main.plex?el=A2QF3CATi4A5FMxe7F5A9ftdqSs5w93hZGi2QHa3g808yQZ>

The ASM Journals program strives for constant improvement in our submission and publication process. Please tell us how we can improve your experience by taking this quick [Author Survey](https://www.surveymonkey.com/r/ASMJournalAuthors).

Sincerely,

Neha Garg

Editor, Microbiology Spectrum
Journals Department
Reviewer comments:

Reviewer #1 (Comments for the Author)

See below

Reviewer #2 (Comments for the Author)

The authors describe the ability of 1-OHPHZ and PYO to kill *P. aeruginosa* delta phz and provide evidence that these molecules are oxidized by ferrous iron that in turn damages bacterial DNA. The data support the conclusions in general.

[#1] Figure 6 of the manuscript is described vaguely and the figure needs additional labeling to follow. Is there another way to demonstrate the oxidized forms of these phenazines? mass spectroscopy?

→ As suggested by the Reviewer, we have added labels to Figure 6 for clarity (**Figure 6**). As mentioned in the manuscript, phenazines are redox-active compounds with distinctive coloration, with two distinct peaks in the UV range and at least one in the visible range, according to the functional groups (Ref. #20 in the manuscript). In this study, we determined the redox states of both 1-OHPHZ and PYO by observing their colors to verify electron transfer between the phenazines and irons (Figure 6). It is also known that oxidized form of PYO is bright blue color with a peak at 278 nm (Ref. #15 in the manuscript). Therefore, we measured UV spectra of oxidized PYO and 1-OHPHZ in the absence/presence of irons in the same conditions to confirm the observation (Supporting information 1). Consistent to the observation (Figure 6B), the UV spectra showed that oxidized PYO (blue) has a peak at ~278 nm, whereas the peak was decreased in PYO partially reduced by ferrous iron (green) (Supporting information 1A). Similar to PYO, we also confirmed the redox state of 1-OHPHZ by both its colors and UV spectra (Supporting information 1B). The supporting information is not included in the manuscript because we believe that the colors of phenazines is sufficient to indicate their redox states.

[#2] Does ferrous iron transfer electrons to 1-OHPHZ and PYO in other media? or are these observations specific to MOPS minimal medium?

→ In our study, we revealed that the phenazine intermediates are generated by electron transfer between both phenazines, 1-OHPHZ and PYO, and ferrous iron, and cause

anaerobic killing of PA14 Δphz using MOPS medium.

We agree with the Reviewer's suggestions regarding the use of other media to determine whether our observations are specific to MOPS media. MOPS is a Zwitterionic biological buffer and is frequently used in many biological and biochemical studies. It is best known as a useful buffering agent for RNA isolation and protein purification (1). Moreover, MOPS is also used as a non-coordinating buffer for solutions containing metal ions because of lack of the ability to form chelates with most metal ions (1).

Autoxidation of transitional metal ions, such as ferrous iron, depends on the reaction conditions, and the rate varies in the presence of different anions. Many studies on metal autoxidation have been performed using unbuffered solutions or different buffers adjusted to various pH. Buffers can affect metal autoxidation (2, 3). Tadolini (1987) reported that ferrous iron is substantially stable for significant time in Mops buffer compared to other buffers. For the reasons, it is widely used in biochemical studies, especially in the study of the redox state of phenazine or iron (4-11). We believe that MOPS medium is most suitable for studying the redox state by electron transfer excluding other influencing factors.

[#3] DNA damage can be measured. Some other assay is needed to confirm that DNA damage has occurred in the PYO and/or 1-OHPHZ and Fe(II) exposed cells besides CFU decrease.

→ We fully agree with the Reviewer's suggestion that DNA damage caused by the phenazine intermediates should be directly measured for definitive conclusions. In this study, we demonstrated that the phenazine intermediates induces DNA damage by measuring the CFU of the *recA* mutant (Figure 8). Since RecA is important for DNA repair in bacteria (Ref. #44), the *recA* mutants are widely used to assess DNA damage (12-15). Thus, we believe that the CFU data of the mutant is sufficient to clearly show that the intermediates cause DNA damage.

In our ongoing follow-up study, we performed a random transposon mutagenesis of PA14 Δphz to elucidate the mechanism of the anaerobic killing. We screened Tn mutants which are resistant to the anaerobic killing condition, compared to a parental strain. Among them, a mutant carries a Tn5 transposon insertion in a putative *ndh* gene encoding NADH dehydrogenase.

CFU data show that the Tn mutant is resistant to the phenazine intermediates (Supporting information 2, unpublished data) as well as two antibiotics, ciprofloxacin and norfloxacin (Supporting information 3, unpublished data). These antibiotics belonging to the class of fluoroquinolone, induce DNA damage by inhibiting DNA gyrase and topoisomerase IV (16, 17). Therefore, our current data support that the phenazine intermediates induce DNA damages (Supporting information 2 & 3, unpublished data).

Minor issues:

[Q1] pg. 2, lines 46-48. The sentence starting with "Otherwise," needs to be rewritten.

→ We have deleted it due to redundancy rather than rewriting the sentence as suggested by the Reviewer (Page 2, Line 52).

[Q2] pg. 2, line 38 "Human" should be "humans".

→ We have revised it as suggested (Page 2, Line 44).

[Q3] pg. 4, line 105 "transferred of" should be "transferred from".

→ We have revised it as suggested (Page 4, Line 108).

[Q4] pg. 4, line 111. "In recent" should be "in recent work".

→ We have revised it as suggested (Page 4, Line 114).

[Q5] pg 4, line 123. "whether it happens" should be "whether electron transfer between the compound and ferrous iron happens".

→ We have revised it as suggested (Page 5, Line 126).

[Q6] pg. 5, line 63. "ability of generation" should be "ability to generate superoxide".

→ We have revised it as suggested (Page 3, Line 67).

[Q7] pg. 3, line 65. "PYO can generate hydroxyl" should be "PYO can generate a hydroxyl".
→ We have revised it as suggested (Page 3, Lines 69–70).

[Q8] pg. 3, line 71. "from the reasons" should be "For these reasons" and moved to the end of the sentence.

→ As suggested by the Reviewer, we have changed from “From the reasons” to “For these reasons”, and moved the sentence to the end of the paragraph (Page 3, Lines 81–82).

[Q9] pg. 3, line 80. "in recent" should be "In recent studies".

→ We have revised it as suggested (Page 3, Line 83).

[Q10] pg. 3, line 90. "It reported" should be "It is reported".

→ As suggested by the Reviewer, we have changed from “It reported” to “It is reported” to make it correct (Page 4, Lines 93–94).

[Q11] pg. 5, lines 125-126. The sentence starting with "In addition" needs to be rewritten.

→ We have revised it as suggested (Page 5, Lines 128–130).

[Q12] pg. 6, line 164. "and catalase whether" should be "catalase to determine whether".

→ We have revised it as suggested (Page 6, Line 168).

[Q13] pg. 6, lines 175-178. Two Sentences starting with "We next" need to be rewritten.

→ We have revised it as suggested (Page 6, Lines 179–182).

[Q14] pg. 6, line 182. Delete "The", capitalize "B" in both.

→ We have revised it as suggested (Page 6, Line 185).

[Q15] pg. 7, line 201. "cell was grown" should be "cells were grown".

→ We have revised it as suggested (Page 7, Line 204).

[Q16] Figure 6 needs additional labeling to follow the Results on page 7 of the text.

→ We have added labels to Figure 6 as suggested (Figure 6).

[Q17] pg. 8, line 234. needs to be rewritten.

→ We have revised it as suggested (Page 8, Line 237).

[Q18] pg. 8, line 242. "In recent" should be "In recent studies".

→ We have revised it as suggested (Page 8, Line 245).

References

1. Garfin DE. 1995. Electrophoretic methods. Introduction to Biophysical Methods for Protein and Nucleic Acid Research, Academic Press, San Diego:53-109.
2. Harris DC, Aisen P. 1973. Facilitation of Fe (II) autoxidation by Fe (III) complexing agents. *Biochimica et Biophysica Acta (BBA)-General Subjects* 329:156-158.
3. Tadolini B. 1987. Iron autoxidation in Mops and Hepes buffers. *Free Radical Research Communications* 4:149-160.
4. Simoska O, Gaffney EM, Lim K, Beaver K, Minteer SD. 2021. Understanding the properties of phenazine mediators that promote extracellular electron transfer in *Escherichia coli*. *Journal of The Electrochemical Society* 168:025503.
5. Jo J, Price-Whelan A, Cornell WC, Dietrich LE. 2020. Interdependency of respiratory metabolism and phenazine-associated physiology in *Pseudomonas aeruginosa* PA14. *Journal of bacteriology* 202:e00700-19.
6. Gubler R, ThomasArrigo LK. 2021. Ferrous iron enhances arsenic sorption and oxidation by non-stoichiometric magnetite and maghemite. *Journal of Hazardous Materials* 402:123425.
7. Wang Y, Wilks JC, Danhorn T, Ramos I, Croal L, Newman DK. 2011. Phenazine-1-carboxylic acid promotes bacterial biofilm development via ferrous iron acquisition. *J Bacteriol* 193:3606-17.
8. Schiessl KT, Hu F, Jo J, Nazia SZ, Wang B, Price-Whelan A, Min W, Dietrich LE. 2019. Phenazine production promotes antibiotic tolerance and metabolic heterogeneity in *Pseudomonas aeruginosa* biofilms. *Nature communications* 10:1-10.
9. Meirelles LA, Newman DK. 2018. Both toxic and beneficial effects of pyocyanin contribute to the lifecycle of *Pseudomonas aeruginosa*. *Molecular microbiology* 110:995-1010.
10. Price-Whelan A, Dietrich LE, Newman DK. 2007. Pyocyanin alters redox homeostasis and carbon flux through central metabolic pathways in *Pseudomonas aeruginosa* PA14. *J Bacteriol* 189:6372-81.
11. Glasser NR, Kern SE, Newman DK. 2014. Phenazine redox cycling enhances anaerobic survival in *Pseudomonas aeruginosa* by facilitating generation of ATP and a proton-motive force. *Molecular microbiology* 92:399-412.
12. Cox MM. 1999. Recombinational DNA repair in bacteria and the RecA protein. *Progress in nucleic acid research and molecular biology* 63:311-366.
13. Aranda J, Bardina C, Beceiro A, Rumbo S, Cabral MP, Barbé J, Bou G. 2011. *Acinetobacter baumannii* RecA protein in repair of DNA damage, antimicrobial resistance, general stress response, and virulence. *Journal of bacteriology* 193:3740-3747.
14. Podlesek Z, Žgur Bertok D. 2020. The DNA damage inducible SOS response is a key player in the generation of bacterial persister cells and population wide tolerance. *Frontiers in microbiology* 11:1785.
15. Chau A, Giang K, Leung M, Tam N. 2008. Role of RecA in the Protection of DNA

- Damage by UV-A in *Escherichia coli*. Journal of Experimental Microbiology and Immunology Vol 12:39-44.
16. Dörr T, Lewis K, Vulić M. 2009. SOS response induces persistence to fluoroquinolones in *Escherichia coli*. PLoS genetics 5:e1000760.
 17. Khodursky AB, Cozzarelli NR. 1998. The mechanism of inhibition of topoisomerase IV by quinolone antibacterials. Journal of Biological Chemistry 273:27668-27677.

October 17, 2022

Prof. Yunho Lee
CHA University
Food Science and Biotechnology
Pocheon, Gyeonggi-do 11160
Korea (South), Republic of

Re: Spectrum02312-22R1 (Pyocyanin and 1-hydroxyphenazine promote anaerobic killing of *Pseudomonas aeruginosa* via single-electron transfer with ferrous iron)

Dear Prof. Yunho Lee:

Your manuscript has been accepted, and I am forwarding it to the ASM Journals Department for publication. You will be notified when your proofs are ready to be viewed.

Sincerely,

Neha Garg
Editor, Microbiology Spectrum
